# EXTEND3D: TOWN-SCALE 3D GENERATION

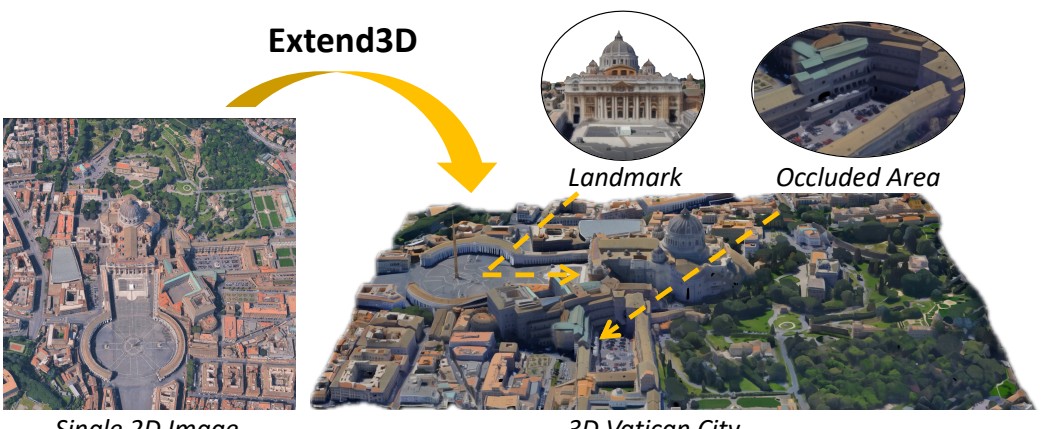

Figure 1: **The large scale result of Extend3D.** We generated large scale ($a = b = 6$) 3D scene from an image of Vatican City captured from Google Earth (Google, 2025).

## ABSTRACT

In this paper, we propose Extend3D, a novel training-free pipeline for 3D scene generation from a single image, built upon an object-centric 3D generative model. To overcome the limitations of fixed-size latent spaces of object-centric models in representing wide scenes, we extend the latent space $(a, b)$ times in $x$ and $y$ directions. Then, by dividing the extended latent into overlapping patches, we utilize the object-centric model on each patch and couple them every time step. In addition, since object-centric models are poor at sub-scene generation, we use the input image and point cloud extracted from a depth estimator as priors to enable this process. Using the point cloud prior, we initialize the structure of the scene and refine the occluded region with iterative under-noised SDEdit. Also, both priors are used to optimize the extended latent during the denoising process so that the denoising paths don't deviate from the sub-scene dynamics. We demonstrate that our method produces better results compared to the previous methods by evaluating human preferences. An ablation study shows that each component of Extend3D has a crucial role in the training-free 3D scene generation.

## 1 INTRODUCTION

In the modern era, 3D scenes are essential in a variety of fields such as games, movies, animations, simulations, and backgrounds for diverse contents. However, it takes a lot of human effort and capital to build decent 3D scenes, even when 3D assets are given. Therefore, end-to-end generative AI for 3D scenes would help reduce such costs and enhance productivity in the industries.

Nevertheless, although recent advances in 3D generative models have enabled automatic creation of high-quality 3D objects, generating large-scale 3D scenes has still been challenging. One reason for the difficulty is that most current 3D datasets used to train conditional 3D generation models (Deitke et al., 2023; Collins et al., 2022; Fu et al., 2021) are curated object-centric, lacking the generative

models to learn cases with complex arrangements of multiple objects and a background. This problem can be solved by collecting or building a dataset containing 3D scene examples; however, such a curation process takes a considerable amount of labor, cost, and computing resources. Consequently, the previous data-centric approach could not enable scene generation. Another reason is that existing latent generative models (Xiang et al., 2025; Team, 2025) represent 3D with a pre-defined size of latent, so that the results are restricted in the level of detail. Due to the limitation in latent size, as the larger or wider 3D scene scales up, the outcome becomes more blurry, similar to an image with confined resolution. To handle this issue and represent enough details of the scene, the latent size should be adaptive to the desired scale of the result.

To address the dataset shortage, research has been conducted to build training-free pipelines that generate 3D scenes using object-centric 3D generative models. A common approach was to generate blocks of a 3D scene in an outpainting manner (Engstler et al., 2025; Zheng et al., 2025). However, the results of that research show that outpainting may degrade the consistency between the blocks, especially in large-scale scenes, and the seams are visible in many cases. Also, while the limited sub-scene generation capability of the object-centric model derives poor results, because outpainting methods utterly rely on the sub-scene generation ability of the pretrained model, the object-centrality of the model cannot or can only be limitedly resolved.

In this paper, we propose **Extend3D**, a novel training-free image-to-3D pipeline for scene generation. To express detailed large-scale 3D scenes, we have extended the latent space of a pretrained image-to-3D object generation model widely, in $x$ and $y$ directions. Due to the arbitrarily extendable latents, our approach is scalable. Motivated by training-free high-resolution image generation works (Bar-Tal et al., 2023; Lee et al., 2023; He et al., 2024; Du et al., 2024; Wu et al., 2025; Lin et al., 2024b;a), which can enhance details in the 2D domain without further training, we divide the extended latent into overlapping patches and generate them simultaneously with the pretrained model. Unlike the previous outpainting approach, which relies on the interaction between denoising paths of patches, this method automatically corrects fine details such as the position, shape, or rotation of objects in the scene.

In addition, we provide the input image and the point cloud extracted from the monocular depth estimator (Wang et al., 2025c) as priors, which are used to initialize and optimize the extended latents. We initialize the structure with the point cloud and refine the occluded region by multiple times of SDEdit (Meng et al., 2022) with the under-noising technique. We also optimize the extended latents every time step to the image and point cloud in order to prevent the denoising paths from deviating from the sub-scene dynamics. Through these priors, Extend3D can overcome the underlying problem of object-centric models in sub-scene generation tasks (e.g., vanishing floor, randomly rotated objects) and overlapped patch generation (e.g., repeated objects, inconsistency in boundary).

The qualitative results show that our method is scalable and generalizable. By human preference evaluation and qualitative experiment, we demonstrate that our method is better in geometry, appearance, and completeness and is faithful to the given image, compared to the state-of-the-art models. With an ablation study, we also prove that overlapping patch-wise flow, initialization, and optimization with priors are essential in training-free 3D scene generation with extended 3D latent.

The main contributions of this paper are:

- We propose a training-free 3D scene generation pipeline based on the 3D object generation model.
- We enabled scalable and detailed 3D scene generation through extended 3D latent.
- We take overlapping patch-wise flow approach to representation local information better.
- We articulate methods to provide priors to the extended latent, especially with under-noising technique.

## 2 RELATED WORK

**3D generative models.** There are many recent studies on generative models that can generate 3D objects from conditions such as texts or images. Currently, their main approach is the latent flow model (Lipman et al., 2022; Rombach et al., 2022) on box-shaped latent or set-based latent (Wiedemann et al., 2025). Trellis (Xiang et al., 2025) generates 3D Gaussian (Kerbl et al., 2023), radiance

field (Mildenhall et al., 2020), and mesh from text or image, using two steps of latent flow models, where each is a box-shaped latent and set-based latent. Hunyuan3D (Team, 2025) utilizes the latent flow model to generate the mesh shape with both box-shaped latent and set-based latent proposed in Zhang et al. (2023a) and inpaint texture via an attention mechanism. TripoSG (Li et al., 2025) also uses the set-based latent of Zhang et al. (2023a) to generate a 3D mesh.

These models have the limitation that they are trained only with object-centric datasets. Moreover, structurally, current flow-based approaches have the problem that their latent size is predefined, so that the output 3D can only have a confined scale of details. We solve these problems by extending the latents so that they can represent a scene.

**Training-free 3D scene generation.** Recent advancement of object-centric 3D generative models and awareness of the 3D scene dataset shortage problem led researchers to try to build training-free 3D scene generation pipelines using object-centric models such as Trellis, Hunyuan3D, or TripoSG.

SynCity (Engstler et al., 2025) generates tiles of 3D sub-scenes separately and sequentially from a text condition. It uses Flux inpainting (Labs, 2024) to generate image conditions with rendered adjacent tiles in 3D, generates a 3D sub-scene with Trellis, and attaches it to 3D. Because SynCity represents a scene as an attachment of separate 3D sub-scenes, even though it blends tiles with additional Trellis denoising processes, there are inconsistencies between tiles, and seams are visible. An image-to-3D scene generation pipeline, 3DTown (Zheng et al., 2025), initializes a scene with the point cloud from VGGT (Wang et al., 2025a) and then completes it patch by patch using the outpainting method proposed in RePaint (Lugmayr et al., 2022) and Trellis. Although 3DTown could generate 3D towns from images in high fidelity, it can only be used with restricted input due to the limitation of object-centric models (e.g., vanishing floor). Also, regardless of the initialization, some objects in the scene ignore some input information, such as rotation, because the information is lost during the reverse process of flow models.

To handle the problems of separate and sequential 3D scene generation, we simultaneously generate 3D sub-scenes with interacting denoising paths instead. With small transitions between overlapping patches, the generation process can effectively capture local information and prevent geometrical errors through simultaneous generation. Also, unlike previous works that rely solely on sub-scene generation using an object-centric model, we optimize the latent representation at each step to prevent paths from transitioning from sub-scene to object dynamics.

**Training-free high-resolution image generation.** In the field of image generation, training-free high-resolution image generation, which has a lot of similarities to extended 3D latent generation, has been widely researched and has led to massive discoveries on the dynamics of the scaled-up latent denoising process. The primary purpose of this area is to generate high-resolution images from pretrained models trained on a relatively low-resolution image dataset. While there are multiple paradigms (He et al., 2024; Shi et al., 2025), we will focus on the overlapping patch-based approach, which is most relevant to our work.

MultiDiffusion (Bar-Tal et al., 2023) generates a high-resolution image from text with an extended 2D latent with overlapping patches, assuming that the model is also trained on cropped images. Each patch has its own denoising path that interacts with other paths in the overlapping region every time step, usually by averaging in the region. Despite the success of MultiDiffusion, especially in panorama image generation, the following research (Du et al., 2024) points out that MultiDiffusion has an object repetition problem, so that it is unable to generate objects with a fixed number or in a desired position. DemoFusion (Du et al., 2024) solves this problem with two ideas: progressive up-sampling and dilated sampling. DemoFusion first generates images on the trained low-resolution model and then upsamples them. While generating a higher-resolution image, with overlapping patches, it mixes upsampled, low-resolution, noised image every time step. DemoFusion also adopts dilated samples, extracted from an extended image by striding pixels in an extended ratio, as well as overlapping patches, to preserve the global structure of the image. Later research, such as CutDiffusion (Lin et al., 2024a) and AccDiffusion (Lin et al., 2024b), additionally refines dilated sampling by randomly permutating pixels across dilated samples in the same position.

When these methods are naively applied to extended 3D latent generation, however, we found that they fail to generate 3D scenes with high fidelity due to the unique dynamics of 3D and object centrality of the model. For example, the floor vanishes or the dynamics between patches are not

correlated well so that there are repeated objects. We therefore provide structure priors to generate a high-fidelity 3D scene, unlike in 2D.

**Generation with priors.** Many studies are trying to provide priors into the pretrained generative model for various purposes. For example, in image editing, they offer an original image to be modified to the intended style as a prior. Another example, additional conditioning, provides conditions such as pose or depth, which is an out-of-scope condition type for text-to-image generation, ensuring that the resulting image also adheres to these conditions.

SDEdit (Meng et al., 2022) is a representative method of image editing that can be applied to differentiable-equation-solving generative models (Ho et al., 2020; Song et al., 2021a; Rombach et al., 2022; Lipman et al., 2022). SDEdit noises the original image (or the corresponding latent) until $t_{start} < 1$ so that it can be partially noised. By denoising the partially noised image from $t_{start}$, SDEdit obtains the edited image as the perturbed distributions of the original image style and the intended image style meet. SDEdit reports that the larger noise induces a more realistic or conditionally aligned result, but reduces faithfulness to the original image.

Zhang et al. (2023b) showed the potential to add an out-of-scope type of condition to a pretrained diffusion model by training a smaller neural network called ControlNet, which is merged with the U-Net in the diffusion model. Readout Guidance (Luo et al., 2024) trains a much smaller neural network called the readout head that can extract properties (e.g., pose, depth, or edge) from the intermediate latent. Then, it calculates the loss with the property and provides the gradient of the loss as guidance, similar to classifier-free guidance (Ho & Salimans, 2021).

We apply SDEdit in our Extend3D to refine the initialized structure. To overcome the trade-off of $t_{start}$, unlike image editing, we propose an under-noising technique designed on the 3D completion task. For optimization, instead of guidance, we optimize the intermediate latent with a loss designed explicitly for 3D scene generation by assuming that the priors have ground truth knowledge on 3D structure and texture, unlike in 2D. In addition, our method eliminates the need for additional training to obtain the loss, unlike 2D methods.

## 3 PRELIMINARIES

### 3.1 LATENT FLOW MODEL FOR 3D GENERATION

A modern approach for a high-quality 3D generative model is latent flow models. They use a box-shaped latent with a fixed size or a set-based latent (e.g. point cloud) in confined region to represent 3D space. Although our approach is not limited to a specific generative model but can be applied to general box-shaped latent or set-based latent flow models, we elaborate our idea on Trellis, which is one of the state of the art 3D generative models.

Trellis generates 3D assets with two steps of latent flow models given an image condition $C_{\mathcal{I}}$ encoded from an image $\mathcal{I}$ by DINOv2 (Oquab et al., 2024), and both steps are generalizable to box-shaped or set-based latent flow model. The first step of the model generates a sparse structure (SS) $\{p_i\} \subset [M]^3$ ($[K] := \{0, 1, ..., K-1\}$), which represents a set of occupied positions in a voxel grid, by denoising box-shaped noise $\mathbf{Z}_1^{SS} \in \mathbb{R}^{N \times N \times N}$ to $\mathbf{Z}_0^{SS} \in \mathbb{R}^{N \times N \times N}$ (usually $M = 4N$) with vector field $v_{SS}$ and decoding it as:

$$\mathbf{Z}_1^{SS} \sim \mathcal{N}(\mathbf{0}, \boldsymbol{I}), \quad \frac{d}{dt}\mathbf{Z}_t^{SS} = \boldsymbol{v}_{SS}(\mathbf{Z}_t^{SS}, C_{\mathcal{I}}, t), \tag{1}$$

$$\{p_i\} = \{p : \mathcal{D}(\mathbf{Z}_0^{SS})_p > 0\}. \tag{2}$$

As the decoder is trained as a VAE, there also exists a trained encoder $\mathcal{E}$ that encodes occupancy voxel $O \in \mathbb{R}^{M \times M \times M}$ to the latent, while it is not used in inference time. The second step of the model conducts denoising on a structured latent (SLat), where a set-based latent feature is matched to a coordinate of sparse structure as:

$$\mathbf{Z}_t^{SLat} = \{(p_i, \mathbf{z}_{i,t})\} \subset [M]^3 \times \mathbb{R}^l, \tag{3}$$

$$\mathbf{z}_{i,1} \overset{iid}{\sim} \mathcal{N}(\mathbf{0}, \boldsymbol{I}), \quad \frac{d}{dt}\mathbf{Z}_t^{SLat} = \boldsymbol{v}_{SLat}(\mathbf{Z}_t^{SLat}, C_{\mathcal{I}}, t), \tag{4}$$

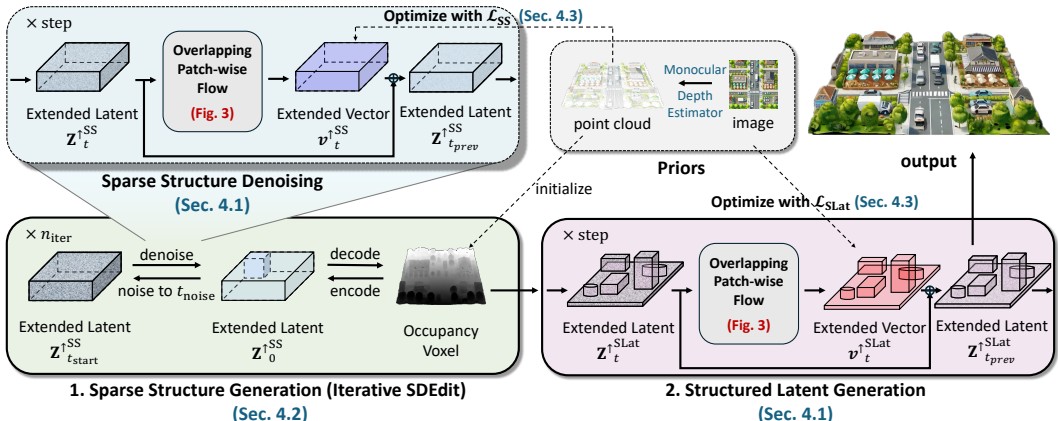

Figure 2: **An overall pipeline of our Extend3D.** Extend3D consists of two parts: sparse structure generation and structured latent generation. In the denoising part of both steps, an overlapping patch-wise flow was used (section 4.1 and fig. 3). In sparse structure generation, iterative SDEdit is used to initialize the structure (section 4.2). Vector fields in both steps are optimized with priors (section 4.3).

with invariant $p_i$ and vector field $v_{\text{SLat}}$. SLat is then decoded to Gaussian, radiance field, or mesh by sparse decoders ($\mathcal{D}_{\text{GS}}$, $\mathcal{D}_{\text{NeRF}}$, and $\mathcal{D}_{\text{mesh}}$). In this paper, we will use the notations $\mathbf{Z}_t$ that can refer to both $\mathbf{Z}_t^{\text{SS}}$ and $\mathbf{Z}_t^{\text{SLat}}$, and $v$ for $v_{\text{SS}}$ and $v_{\text{SLat}}$ for simplicity.

## 3.2 SDEDIT

We introduce SDEdit since we use it to refine the initialized structure, considering it as a 3D editing task. SDEdit noises latent of a "guide" (e.g., image to be edited) $\mathbf{Z}_0^{(g)}$ to $\mathbf{Z}_{t_{\text{start}}}$ and denoises it to $\mathbf{Z}_0$ to get the edited result. With the added noise, the perturbed distribution meets the intended distribution while preserving information in the guidance. Originally, SDEdit was designed for diffusion models (Song et al., 2021b), but it can also be used in flow models, with the following equations:

$$\mathbf{Z}_{t_{\text{start}}} = (1 - t_{\text{start}}) \cdot \mathbf{Z}_0^{(g)} + t_{\text{start}} \cdot \epsilon, \quad \epsilon \sim \mathcal{N}(\mathbf{0}, \boldsymbol{I}), \tag{5}$$

$$\frac{d}{dt}\mathbf{Z}_t = \boldsymbol{v}(\mathbf{Z}_t, C, t), \tag{6}$$

where $C$ refers to the editing condition. When $t_{\text{start}}$ increases, the denoising path gets longer, causing the effect of conditioning and generative models to be enlarged.

## 4 METHOD

Extend3D is a training-free pipeline that generates a 3D scene from a single scene image. To implement Extend3D, we extended 3D latents of a pretrained object-centric 3D generative model (Xiang et al., 2025) so that it can represent more detailed and larger 3D scenes. We extend the latents in the $x$ and $y$ coordinates, and a part of the extended latent works as a usual latent for the pretrained object-centric model.

To handle extended latents, we divide them into overlapping patches that are generated simultaneously with separate but coupled denoising paths conditioned on image patches (section 4.1). Additionally, to address the underlying issues of the object-centric model (e.g., vanishing floor, inability to generate sub-scenes, and randomly rotated objects), and to mitigate the problems associated with patch-wise generation (e.g., repeated objects and seams between patches), we incorporate priors into the generation process. We first initialize the scene with a point cloud from the depth estimator and perform iterative SDEdit to complete the occluded area and refine the scene while generating the structure (section 4.2). We then optimize the scene every time step with a point cloud for the structure and an image for the entire scene. We also propose a loss function particularly designed to give a point cloud as a prior to the voxel (section 4.3). The overall pipeline is illustrated in fig. 2 and appendix A.1.

Figure 3: **Overlapping patch-wise flow.** The extended latent is divided into original size latent patches with the sliding window. Then we get the patch vector from each latent patch and merge them into a single extended latent vector so that the patches are coupled.

## 4.1 OVERLAPPING PATCH-WISE FLOW

In order to generate detailed 3D structure and texture, we introduce extended latents $\mathbf{Z}^{\uparrow \mathrm{SS}}_t \in \mathbb{R}^{aN \times bN \times N}$ and $\mathbf{Z}^{\uparrow \mathrm{SLat}}_t \subset [aM] \times [bM] \times [M] \times \mathbb{R}^l$ ($a$ and $b$ are extension factor) where $\mathbf{Z}^{\uparrow}_t$ can refer to both. We divide these latents into $((a-1)d+1) \times ((b-1)d+1)$ overlapping patches with division factor $d$. The $(i,j)$ latent patches are defined as:

$$\mathbb{W}^K_{i,j} = \left[\frac{iK}{d}, \frac{iK}{d}+K\right) \times \left[\frac{jK}{d}, \frac{jK}{d}+K\right) \times [K], \tag{7}$$

$$\phi^{\mathrm{SS}}_{i,j}(\mathbf{Z}^{\uparrow \mathrm{SS}}_t) = (\mathbf{Z}^{\uparrow \mathrm{SS}}_t)_{\mathbb{W}^N_{i,j}} \in \mathbb{R}^{N \times N \times N}, \tag{8}$$

$$\phi^{\mathrm{SLat}}_{i,j}(\mathbf{Z}^{\uparrow \mathrm{SLat}}_t) = \left\{ \left(\boldsymbol{p} - \left(\frac{iM}{d}, \frac{jM}{d}, 0\right), \mathbf{z}_t\right) : (\boldsymbol{p}, \mathbf{z}_t) \in \mathbf{Z}^{\uparrow \mathrm{SLat}}_t, \boldsymbol{p} \in \mathbb{W}^M_{i,j} \right\}. \tag{9}$$

The subtraction in eq. (9) is for coordinate normalization into $[M]^3$. This process can be described as a $N^3$ or $M^3$ sized sliding window $W$ moving with stride $N/d$ or $M/d$ to sample patches, illustrated in fig. 3. The patches can be mapped to their original position by:

$$(\phi^{\mathrm{SS}}_{i,j})^{-1}(\boldsymbol{X})_{x,y,z} = \mathbf{1}_{(x,y,z) \in \mathbb{W}^N_{i,j}} \cdot \boldsymbol{X}_{x-\frac{iN}{d}, y-\frac{jN}{d}, z}, \tag{10}$$

$$(\phi^{\mathrm{SLat}}_{i,j})^{-1}(\boldsymbol{X}) = \left(\boldsymbol{X} + \left(\left(\frac{iM}{d}, \frac{jM}{d}, 0\right), \mathbf{0}\right)\right) \cup \{(\boldsymbol{p}, \mathbf{0}) : \boldsymbol{p} \in \{\boldsymbol{p}_i\}, \forall \mathbf{z} \, (\boldsymbol{p}, \mathbf{z}) \notin X\}, \tag{11}$$

setting the value of other positions to zero. We also patchified the image condition with $\psi_{i,j}$ that can crop the image region that exactly matches the 3D patch $(i,j)$ (see details in appendix A.2). Similar to MultiDiffusion, we get the vector field of the extended latents by merging the vector fields for each patch, where the overlapping regions are averaged across the patches, as illustrated in fig. 3. The entire overlapping patch sampling, merging, and denoising process can be formulated as:

$$\boldsymbol{v}^{\uparrow}(\mathbf{Z}^{\uparrow}_t, \mathcal{I}, t) = \sum_{i,j} \phi^{-1}_{i,j}(\boldsymbol{v}(\phi_{i,j}(\mathbf{Z}^{\uparrow}_t), C_{\psi_{i,j}(\mathcal{I})}, t)) \oslash \sum_{i,j} \mathbf{1}_{\mathbb{W}_{i,j}}, \tag{12}$$

$$\frac{d}{dt}\mathbf{Z}^{\uparrow}_t = \boldsymbol{v}^{\uparrow}(\mathbf{Z}^{\uparrow}_t, \mathcal{I}, t). \tag{13}$$

The term in the summation can be calculated independently from the other patches, but the dynamics of different patches, even far away, can be coupled by overlapping regions.

Divided but coupled dynamics can refine errors in other patches. By the small movement of the sliding window, our method can detect local information from the change in image and latent between patches. Also, because some objects are in the center of some patches, we can leverage the object-centric model better. The beneficial effect of overlapping patch-wise flow can be verified by changing $d$ as shown in fig. 5 (A).

Noted in DemoFusion (Du et al., 2024), AccDiffusion (Lin et al., 2024b), and CutDiffusion (Lin et al., 2024a), dilated sampling is crucial for generating a consistent global structure. We apply dilated sampling in the sparse structure generation phase and leave the details in appendix A.3.

## 4.2 INITIALIZE WITH PRIOR

If the denoising process of sparse structure starts from perfect Gaussian noise and denoised with eq. (13), all patches fail to initialize the sub-scene by the limitation of object-centric model. Moreover, the low frequency of the structure is determined in the early time step (Wu et al., 2025) before the patches are coupled well. Consequently, the output structure is very noisy and broken, as in fig. 5 (B). Therefore, it is necessary to initialize the scene structure with a prior.

Motivated by 3DTown (Zheng et al., 2025), we initialize the scene structure with a point cloud $\mathbb{P}$ extracted from a monocular depth estimator. We adopted MoGe-2 (Wang et al., 2025b;c) for our Extend3D. The point cloud is voxelized to an occupancy voxel grid $\boldsymbol{O}^{\uparrow}{}_0 \in \mathbb{R}^{aM \times bM \times M}$. Since the occupancy voxel from the depth estimator cannot represent the occluded area of 3D, this empty region should be rectified using the pretrained model. To solve this problem, with $\mathbf{Z}^{\uparrow}{}_0^{(g)} = \mathcal{E}(\boldsymbol{O}^{\uparrow}{}_0)$, our Extend3D performs SDEdit. Unlike standard SDEdit, instead of eq. (5), we noise the guidance structure by:

$$\mathbf{Z}^{\uparrow}{}_{t_{\text{start}}} = (1 - t_{\text{noise}}) \cdot \mathbf{Z}^{\uparrow}{}_0^{(g)} + t_{\text{noise}} \cdot \epsilon, \quad \epsilon \sim \mathcal{N}(\mathbf{0}, \boldsymbol{I}), \tag{14}$$

with $t_{\text{start}} > t_{\text{noise}}$, so the latent is denoised more than how it was noised. By under-noising, the undesired empty area might be considered as an extra noise to the model, so that it can be filled in the denoising process, similar to Jeong et al. (2025) applying noise on the high-frequency region for detail refinement in the image domain. We prove the effectiveness of this choice in section 5.4.

SDEdit can fill the undesired empty area; however, it often fails to complete the scene fully, and some holes remain. To mitigate this, as a single SDEdit process partially refines the structure, we apply SDEdit multiple times (usually 5 times) as: $\boldsymbol{O}^{\uparrow}{}_n = \text{SDEdit}(\boldsymbol{O}^{\uparrow}{}_{n-1})$. With this process, the proposed Extend3D can iteratively fill the occluded region of $\boldsymbol{O}^{\uparrow}{}_0$, eventually completing the scene.

## 4.3 OPTIMIZE WITH PRIOR

During the denoising process, sub-scenes deviate from scene-like structure to object-like structure, by the properties of the object-centric model, causing distortion or a vanished floor, even though proper initialization was used. To prevent deviation and drag the denoising paths to the paths that follow the condition appropriately, we optimize the extended latent between time steps with the point cloud and image. When solving eq. (13) with the discrete ODE solver, instead of moving directly along $\boldsymbol{v}^{\uparrow}(\mathbf{Z}^{\uparrow}{}_t, \mathcal{I}, t)$, we use $\hat{\boldsymbol{v}}_t^{\uparrow}$, an optimized vector starting from $\boldsymbol{v}^{\uparrow}(\mathbf{Z}^{\uparrow}{}_t, \mathcal{I}, t)$. By optimizing the vector field, we can utilize the pretrained model for the occluded region while optimizing on the seen region, like Readout Guidance. In addition, optimizing the vector field can bring more consistency between patches, since the entire scene is optimized at once, as in Lee et al. (2023).

In the sparse structure generation process, we use the loss:

$$\mathcal{L}_{\text{SS}} = -\frac{1}{|\mathbb{P}|} \sum_{\boldsymbol{p} \in \mathbb{P}} \log \sigma((\mathcal{D}(\mathbf{Z}^{\uparrow}{}_t^{\text{SS}} - t \cdot \hat{\boldsymbol{v}}_t^{\uparrow}))_{\boldsymbol{p}}), \tag{15}$$

where $\sigma$ is a sigmoid function. The loss function is designed to enforce that the initialized voxels do not disappear during the denoising process, motivated by binary cross-entropy loss. It gives a positive signal on predicted voxels where points exist. Voxels with dense point clouds will have more weight in the loss. While this loss can be minimized by increasing the voxel values, combined with the pretrained model, it merely prevents the desired voxels from disappearing, rather than creating undesired voxels. With the $\mathcal{L}_{SS}$, we optimize $\hat{\boldsymbol{v}}_t^{\uparrow}$ with Adam optimizer (Kingma & Ba, 2015).

In the structured latent generation process, we use the loss (Zhang et al., 2018; Wang et al., 2004):

$$\mathcal{L}_{\text{SLat}} = \text{LPIPS}(\hat{\mathcal{I}}, \mathcal{I}) - \text{SSIM}(\hat{\mathcal{I}}, \mathcal{I}), \quad \hat{\mathcal{I}} = \text{Render}(\mathcal{D}_{\text{GS}}(\mathbf{Z}^{\uparrow}{}_t^{\text{SLat}} - t \cdot \hat{\boldsymbol{v}}^{\uparrow}), \boldsymbol{P}), \tag{16}$$

where $\text{Render}$ is a differentiable renderer for Gaussian splatting and $\boldsymbol{P}$ is a camera parameter of an image viewpoint provided by the depth estimator. This optimizes the entire scene with an image in the original camera view. Because an object-centric model often loses details in the texture of the scene, this optimization helps to refine it. Also, it makes the seams between patches invisible since the boundary is also optimized every time step, so that the paths can be more consistent with each other. With $\mathcal{L}_{\text{SLat}}$, we also optimize $\hat{\boldsymbol{v}}_t^{\uparrow}$ with Adam.

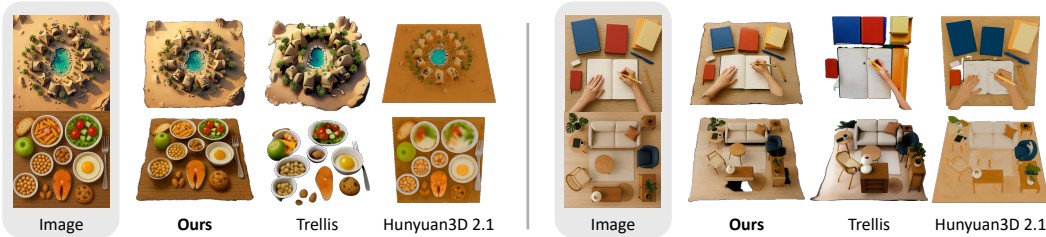

Figure 4: **The results of our Extend3D.** Our 3D scene generation results (with extension factors $a = b = 2$) are compared to the results of state-of-the-art 3D generative models. While previous methods may not accurately represent the image or lose scene details, our method effectively expresses the image condition in 3D. All the input images are AI-generated from ChatGPT (OpenAI, 2023) or Flux.1 [dev] (Labs, 2024). There are more results in appendix A.4

## 5 EXPERIMENTS

| | Geometry | Faithfulness | Appearance | Completeness |
|---|---|---|---|---|
| vs Trellis | 77.6 | 81.9 | 74.3 | 79.5 |
| vs Hunyuan | 89.0 | 91.4 | 91.4 | 85.2 |

Table 1: Human preference win rate (%) of our method.

| | LPIPS ↓ | SSIM ↑ | PSNR ↑ |
|---|---|---|---|
| Trellis | 0.825 | 0.216 | 9.59 |
| Hunyuan | 0.856 | 0.271 | 10.8 |
| **Ours** | **0.439** | **0.469** | **17.6** |

Table 2: Qualitative results.

### 5.1 HUMAN PREFERENCE

We evaluated the performance of our Extend3D through human preference since there is no image-3D-scene pair dataset for data-centric evaluation. We compared our method with Trellis and Hunyuan3D-2.1, which are the current best open source 3D generation models. Human annotators (15 participants) ranked three methods on four criteria: geometry, faithfulness, appearance, and completeness, provided 14 image and 3D scene sets. As a result (table 1), our Extend3D outperformed previous methods in four criteria.

### 5.2 QUANTITATIVE RESULTS

We render the 3D scene to the camera view of the input image get from the camera parameter estimator (Wang et al., 2025c) and get LPIPS, SSIM, and PSNR score with 18 input images, which contains diverse and wide scenes. As in table 2, our method scored the best scores in three metrics, which means that it is most faithful to the input image in structure and texture quantitatively.

### 5.3 QUALITATIVE RESULTS

Fig. 1 (We didn't use SLat optimization in large scale generation due to the memory shortage) shows the scalability of our method. Given a town-scale scene image, the extended latent could fully express the details, including the landmarks and small buildings, and output a 36 times larger result compared to the original latent space. We present more examples of wide scenes in appendix A.4. Because our Extend3D was built upon a general object-centric 3D model, our method is also generalizable as in fig. 4. It can generate a town, a table of foods, a scene of studying, and an indoor room. In diverse cases, our method outperformed previous 3D generative models. However, there is a limitation that, in some cases, the occluded region is not perfectly filled. The last example (the one representing a room) of fig. 4 shows the fallacy in the structure completion.

### 5.4 ABLATION STUDY

We conducted an ablation study on three proposed methods in our Extend3D, and represented it in fig. 5. When we obtained the ablation results for fig. 5 (A) and (B), we did not optimize the latent to emphasize the structural difference in camera view.

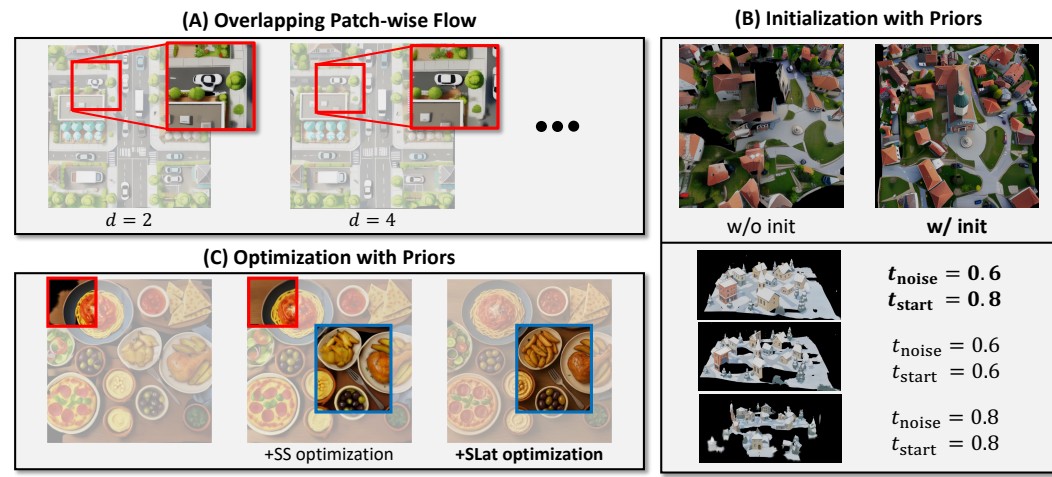

Figure 5: **Ablation study.** All the images, except for the ablation of under-noising, are taken from the input image camera viewpoint. We set $a = b = 2$ to generate the 3D scenes in this figure.

**Overlapping Patch-wise Flow.** We claim that the coupled paths of patches can rectify each other and catch local information well. To validate this argument, we compare the results from the varying division factor $d$. As illustrated in fig. 5 (A), $d = 2$ caused distortion in local structure while $d = 4$ did not. These results demonstrate that the interaction between patches corrects each other, and the stride of the sliding window enables the extended latent dynamics to understand finer details.

**Initialize with Prior.** In the first part of fig. 5 (B), the results with and without initialization are compared. Without initialization (i.e. $t_{start} = 1$), the structure is totally broken with important objects disappeared, buildings not in proper position, etc. We therefore conclude that proper initialization is essential for extended latents. In the second part, we compared three results with different $t_{noise}$ and $t_{start}$. When $t_{noise} = t_{noise}$ (usual SDEdit), the structure maintained the holes in the initial point cloud or totally destroyed due to the $t_{start}$ trade-off relationship in SDEdit. However, with $t_{noise} < t_{start}$ (under-noising), the occluded region of the initial structure are completed naturally.

**Optimize with Prior.** Fig. 5 (C) shows the ablation study for optimization with priors. Starting from not optimizing, we added sparse structure optimization and structured latent optimization sequentially. Without sparse structure optimization, the floor and some parts of the objects vanished as a result of the object-centric models in fig. 4. Structured latent optimization could refine seams and distortion between patches compared to those without optimization. Furthermore, overall quality in the structure and texture of the scene enhanced, for example, the fork and chips in the figure.

## 6 CONCLUSION

We proposed a training-free 3D scene generation pipeline, Extend3D. By the extended latent from the pretrained object-centric model, we enabled scalable 3D scene generation. Our Extend3D produced general and promising 3D scenes. It also outperformed state-of-the-art 3D generative models. We demonstrated that the methods proposed in our work (overlapping patch-wise flow, initialization with priors, and optimization with priors) and techniques in such methods (iterative SDEdit, under-noising, and the loss functions) made a substantial positive effect on 3D scene generation. Our method is also applicable to general object-centric, pre-trained flow models that use box-shaped or set-based latent representations. We expect that the evolution of object-centric 3D generative models would enable better 3D scene generation through our approach.

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

## A APPENDIX

### A.1 ALGORITHMS

**Algorithm 1** Sparse Structure Generation

1: **Input**: $\mathcal{I}, \mathbb{P}$
2:
3: $\boldsymbol{O}^{\uparrow}{}_0 \leftarrow \mathbf{1}_{\mathbb{P}}$
4: Define schedule
5: $\quad [t_{\text{start}} = t_1 > ... > t_k = 0]$
6: **for** $0 \leq n < n_{\text{iter}}$ **do**
7: $\quad \mathbf{Z}^{\uparrow(g)}_0 \leftarrow \mathcal{E}(\boldsymbol{O}^{\uparrow}{}_n)$
8: $\quad$ Sample $\epsilon \sim \mathcal{N}(\mathbf{0}, \boldsymbol{I})$
9: $\quad \mathbf{Z}^{\uparrow}{}_{t_1} \leftarrow (1 - t_{\text{noise}}) \cdot \mathbf{Z}^{\uparrow(g)}_0 + t_{\text{noise}} \cdot \epsilon$
10: $\quad$ **for** $1 \leq m < k$ **do**
11: $\quad\quad$ Initialize $\hat{\boldsymbol{v}}^{\uparrow} \leftarrow \boldsymbol{v}^{\uparrow}(\mathbf{Z}^{\uparrow}{}_{t_m}, \mathcal{I}, t_m)$
12: $\quad\quad$ Optimize $\hat{\boldsymbol{v}}^{\uparrow}$ with Adam
13: $\quad\quad \mathbf{Z}^{\uparrow}{}_{t_{m+1}} \leftarrow \mathbf{Z}^{\uparrow}{}_{t_m} + (t_{m+1} - t_m) \cdot \hat{\boldsymbol{v}}^{\uparrow}$
14: $\quad$ **end for**
15: $\quad \boldsymbol{O}^{\uparrow}{}_{n+1} \leftarrow (\mathcal{D}(\mathbf{Z}^{\uparrow}{}_0) > 0)$
16: **end for**
17: **return** $\{\boldsymbol{p} : (\boldsymbol{O}^{\uparrow}{}_{n_{\text{iter}}})_{\boldsymbol{p}} > 0\}$
18:

**Algorithm 2** Structured Latent Generation

1: **Input**: $\{p_i\}, \mathcal{I}, \boldsymbol{P}$
2:
3: Initialize $\mathbf{Z}^{\uparrow}{}_1$ with $\{\boldsymbol{p}_i\}$
4: Define schedule $[1 = t_1 > ... > t_k = 0]$
5: **for** $1 \leq m < k$ **do**
6: $\quad$ Initialize $\hat{\boldsymbol{v}}^{\uparrow} \leftarrow \boldsymbol{v}^{\uparrow}(\mathbf{Z}^{\uparrow}{}_{t_m}, \mathcal{I}, t_m)$
7: $\quad$ Optimize $\hat{\boldsymbol{v}}^{\uparrow}$ with Adam
8: $\quad \mathbf{Z}^{\uparrow}{}_{t_{m+1}} \leftarrow \mathbf{Z}^{\uparrow}{}_{t_m} + (t_{m+1} - t_m) \cdot \hat{\boldsymbol{v}}^{\uparrow}$
9: **end for**
10: **return** $\mathbf{Z}^{\uparrow}{}_0$

**Algorithm 3** Extend3D

1: **Input**: $\mathcal{I}$
2:
3: $\mathbb{P}, \boldsymbol{P} \leftarrow MoGe2(\mathcal{I})$
4: $\{\boldsymbol{p}_i\} \leftarrow SS(\mathcal{I}, \mathbb{P})$
5: $\mathbf{Z}^{\uparrow} \leftarrow SLat(\{\boldsymbol{p}_i\}, \mathcal{I}, \boldsymbol{P})$
6: **return** $(\mathcal{D}_{\text{GS}}, \mathcal{D}_{\text{NeRF}}, \mathcal{D}_{\text{mesh}})(\mathbf{Z}^{\uparrow})$

### A.2 IMAGE PATCHIFICATION

We patchify an input image precisely using the coordinates of the point cloud extracted from the monocular depth estimator. We can map a pixel $\mathcal{I}_{x,y}$ to a coordinate of the point cloud $q_{x,y}$ in the extended latent space. When $q_{x,y} \in \mathbb{W}_{i,j}$, the 3D area corresponding to the given pixel is in the patch $(i, j)$ since the structure was initialized with the depth estimator. We therefore define the image patch $\psi_{i,j}(\mathcal{I})$ by collecting all $\mathcal{I}_{x,y}$ where corresponding $q_{x,y} \in \mathbb{W}_{i,j}$, setting the other pixels to be black, and cropping out black regions so that the image becomes square.

### A.3 DILATED SAMPLING

We follow the recipe for dilated sampling in Lin et al. (2024b). In this section, we assume that the unextended latent shape is $K \times K \times K$ for the sake of generalization. We divide the extended latent into $K \times K$ non-overlapping patches so that the size of each patch is $a \times b \times K$. We then randomly sample a pillar of 3D latent in each patch. The sampled pillars are attached by maintaining their relative positions to be a $K \times K \times K$ shaped latent. We sample $a \times b$ samples without replacement, and we call them dilated samples. The dilated samples pass through the pretrained model with image condition $C_{\mathcal{I}}$ without image patchfication. When dilated sampling is applied, eq. (12) is altered to be:

$$\boldsymbol{v}^{\uparrow}(\mathbf{Z}^{\uparrow}{}_t, \mathcal{I}, t) = (1 - \gamma_t)\text{PatchWise}(\mathbf{Z}^{\uparrow}{}_t, C_{\mathcal{I}}, t) + \gamma_t\text{Dilated}(\mathbf{Z}^{\uparrow}{}_t, C_{\mathcal{I}}, t), \quad (17)$$

$$\gamma_t = 0.5 \cos^{\alpha}(\pi - \pi t) + 0.5, \quad (18)$$

where $\text{PatchWise}$ is equal to eq. (12) and $\alpha$ is a hyperparameter. We set $\alpha = 5$ in our experiments. We use dilated sampling only in the sparse structure generation because we found that dilated sampling worsens the texture when applied to structured latent generation empirically.

### A.4 MORE RESULTS

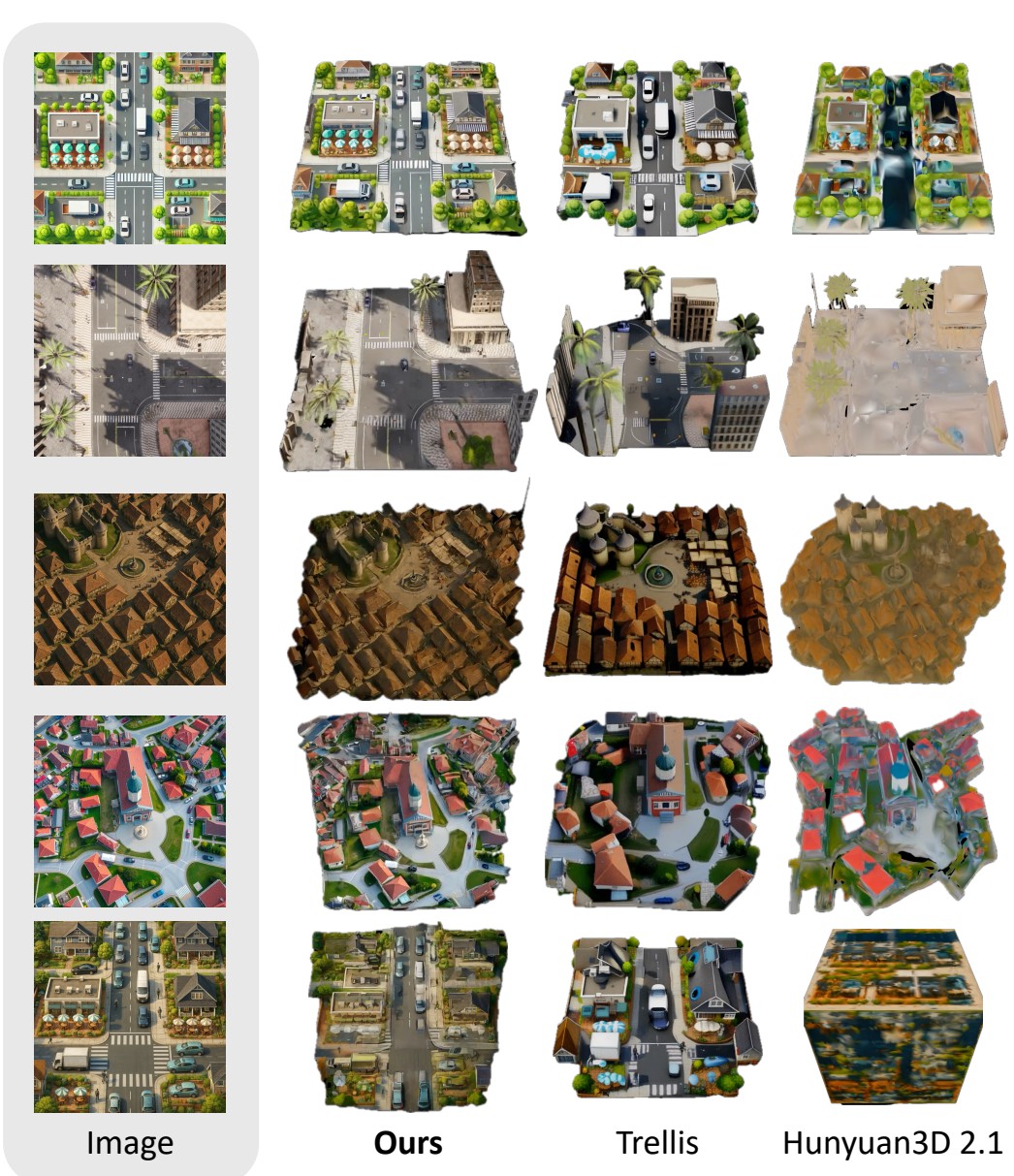

Image     **Ours**     Trellis     Hunyuan3D 2.1

Figure 6: **Example results for diverse images.** In this figure, we set $a = b = 2$. We compared our results with the state-of-the-art open source 3D generative models. The second image is from CarlaSC dataset (Wilson et al., 2022).The other images are generated by ChatGPT (OpenAI, 2023) and Flux.1 [dev] (Labs, 2024).

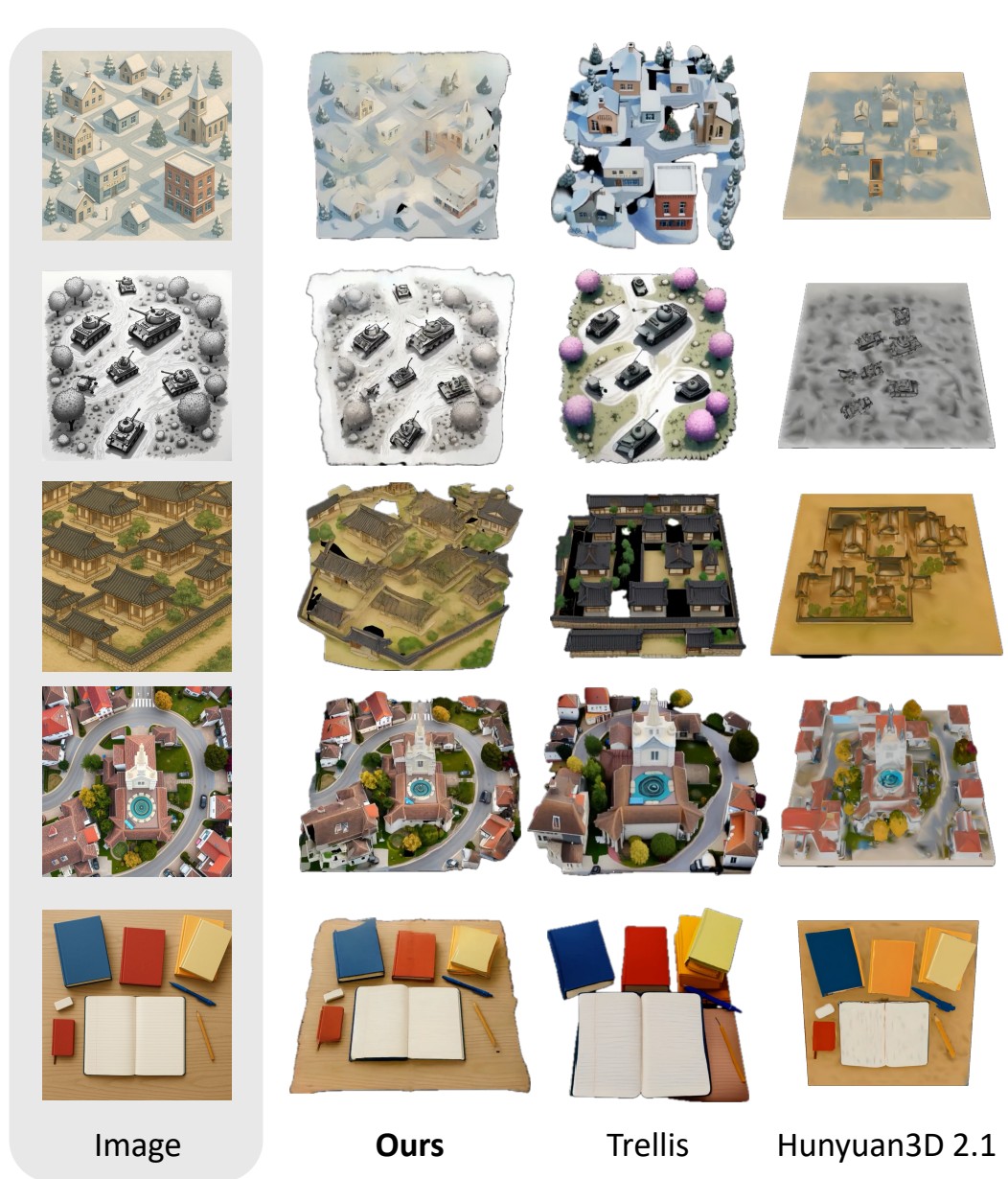

Figure 7: **Example results for diverse images.** In this figure, we set $a = b = 2$. We compared our results with the state-of-the-art open source 3D generative models. The images are generated by ChatGPT (OpenAI, 2023) and Flux.1 [dev] (Labs, 2024).

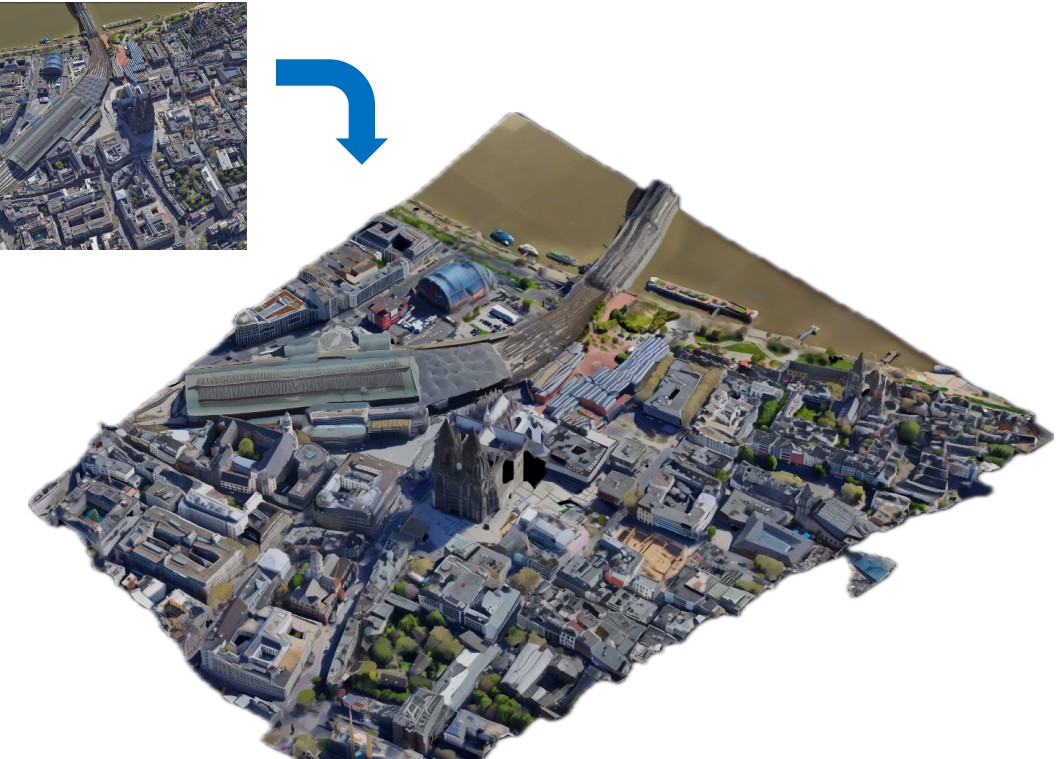

Figure 8: **The large scale result of Extend3D.** We generated large scale ($a = b = 6$) 3D scene from the image of Köln captured from Google Earth (Google, 2025). We didn't use SLat optimization in this result due to the memory shortage.

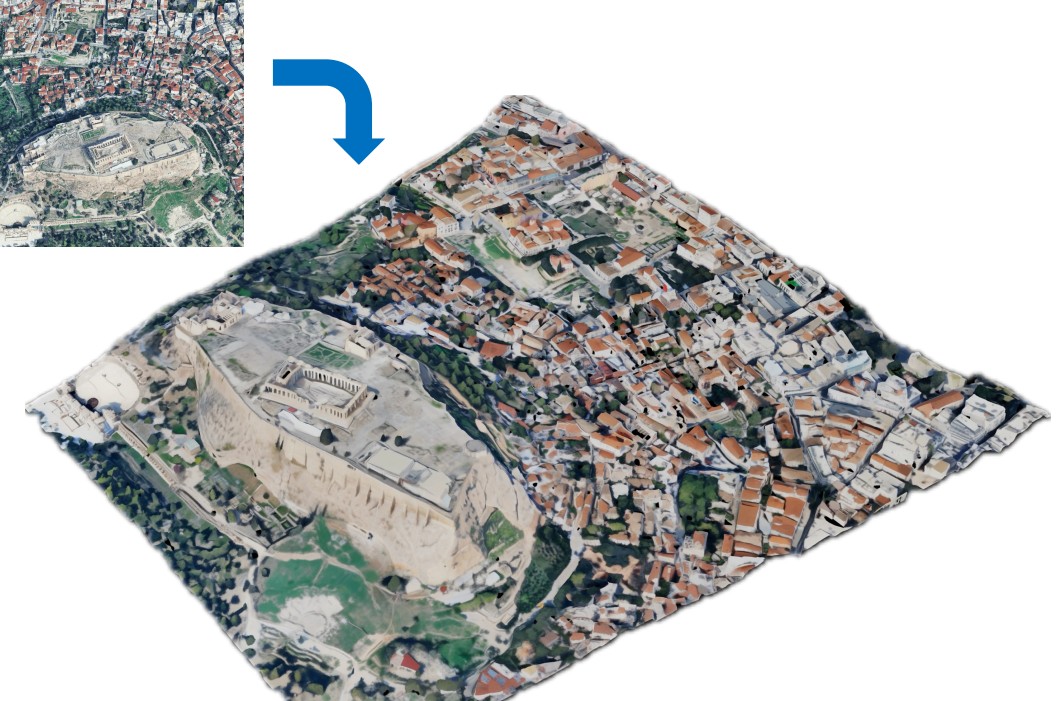

Figure 9: **The large scale result of Extend3D.** We generated large scale ($a = b = 6$) 3D scene from the image of Athens captured from Google Earth (Google, 2025). We didn't use SLat optimization in this result due to the memory shortage.

## A.5 LARGE LANGUAGE MODELS

We used large language models when polishing the paper, and as a coding assistant.