# OpenReview forum: "Extend3D: Town-scale 3D Generation"
_ICLR.cc/2026/Conference — ICLR 2026 Conference Withdrawn Submission_

### Official Review · Reviewer_1oPL · 2025-10-20

**Soundness:** 2
**Presentation:** 2
**Contribution:** 2
**Rating:** 2
**Confidence:** 5

**Summary:**

This paper proposes a method for 3D scene reconstruction from near-top-view images of a scene. The approach uses depth estimation to obtain an initial point cloud. The image is then divided into overlapping patches, which guide the generation of multiple overlapping trellis blocks for large-scale scene modeling.

**Strengths:**

- the idea of using global point cloud from depth estimator to guid the local scene generation make sense

**Weaknesses:**

- The method assumes a top-view perspective and near-isometric conditions, such that the patchified images can align with the 3D latent windows.  Can this approach handle viewpoint change?
- The paper primarily showcases textured mesh comparisons, leaving the geometric quality compared to the trellis baseline ambiguous. Visualization of untextured geometry (e.g., in Figure 6) is highly recommended to better evaluate structural accuracy.
- Comparisons with scene-level shape generation works, such as SyncCity, are missing.
- Important citations are absent, e.g. BlockFusion (Wu et al., BlockFusion: Expandable 3D Scene Generation using Latent Tri-plane Extrapolation).
- What is the max number of patches that can be generated simultaneously?

**Questions:**

see above

---

### Official Review · Reviewer_utD6 · 2025-10-29

**Soundness:** 2
**Presentation:** 2
**Contribution:** 2
**Rating:** 2
**Confidence:** 4

**Summary:**

This work introduces Extend3D, a training-free pipeline for generating large-scale 3D scenes from a single image. Extend3D builds upon existing object-centric 3D generative models and employs a patch-based generation approach with overlapping patch-wise flow, and depth and image priors informed optimization. Experiments show that Extend3D outperforms existing 3D generative models.

**Strengths:**

- The proposed pipeline does not require training on large-scale 3D scene datasets by repurposing existing object-centric 3D generative models (Trellis).
- Extend3D addresses the limitations of object-centric 3D generative models by extending the latent space, dividing it into overlapping patches, and jointly denoising them. It further guides the generation process with depth and image priors and performs iterative SDEdit refinement.

**Weaknesses:**

- The idea of applying object-centric generative models to large 3D scene generation has been explored in prior works (e.g., 3DTown [Zheng et al. 2025]). The overlapping patch-wise flow extends multidiffusion [Bar-Tal et al. 2023]. And the depth guided generation is similar to 3DTown. So the overall technical novelty is limited.
- The experimental evaluation is limited to human preference studies. The authors argue that there is no iamge-3D-scene pair dataset for quantitative evaluation, but there are existing datasets like UrbanScene3D [Lin et al. 2022] or large synthetic 3D scene models online (e.g., sketchfab.com). I would expect evaluation metrics measuring 3D geometry generation quality as well as 2D visual quality measurement in novel views. The ablation study is also limited to qualitative results.
- Baseline comparisons include only object-centric 3D generative models. Recent training-free 3D scene generation methods like SyncCity [Engstler et al. 2025] is not included.
- The writing needs to be improved for clarity. For example, in Sec 4.1, the equations do not help much in explaining the simple idea of patchfication and overlapping patch-wise flow. The contribution bullet points are vague and not specific to this work.
- In related work, a subsection on 3D scene generation (works like outdoor scene generation NuiScene [Lee et al. 2025], Pyramid Discrete Diffusion [Liu et al. 2024], and indoor scene generation BlockFusion [Wu et al. 2024], LT3SD [Meng et al. 2025]) is worth considering.

**Questions:**

- Can the authors provide quantitative evaluation and ablation on existing 3D scene datasets?
- Can existing training-free 3D scene generation baselines be compared qualitatively and quantitatively?
- Some generation results seem to deviate from input images, for example, in Fig 7, rows 1 and 3. The image-3D consistency seems concerning.
- In L419, the authors mention that they did not include certain optimization due to computational constraints. However, running time and memory usage were not discussed in the paper.

---

### Official Review · Reviewer_n7vC · 2025-10-31

**Soundness:** 3
**Presentation:** 2
**Contribution:** 3
**Rating:** 4
**Confidence:** 4

**Summary:**

This paper presents Extend3D, a training-free framework for image-conditioned large-scale 3D scene generation, built upon the object-level pretrained 3D generative model Trellis. Unlike prior works, Extend3D does not require collecting or training on scene-level 3D datasets.

The paper highlights three main contributions:
1. It introduces an overlapping patch-wise flow strategy that performs diffusion process simultaneously across multiple overlapping patches extracted from the input image. Consistency across overlapping regions is maintained via averaging.
2. It leverages MoGe-2 to estimate a point cloud for initialization, obtaining the initial latent representation. Noise is added for $t_{noise}$ steps, followed by denoising steps beyond $t_{noise}$, encouraging the model to fill in potentially empty regions.
3. It proposes a flow optimization scheme for Trellis, ensuring that generated 3D structures remain present at initialized point locations and that the rendered 3D scene aligns well with the input image.

The method is compared with object-centric models Trellis and Hunyuan3D 2.1, demonstrating superior results. Ablation studies further validate the effectiveness of each proposed component.

**Strengths:**

1. Training-free framework: The approach does not require scene-level 3D data, instead extending object-level pretrained models to the scene generation domain.
2. Novel SDEdit variant: The proposed SDEdit-based denoising strategy is interesting—it relies on stronger denoising than noise injection, encouraging the model to treat empty regions as noise and thus fill them more effectively.
3. Effective flow optimization: The flow refinement process significantly improves scene completeness and alignment with the input image, as supported by the ablation results.

**Weaknesses:**

1. The validation dataset is insufficiently described, and using only 18 images for evaluation seems limited (e.g., 3DTown employs 100 images as a benchmark).
2. The method lacks comparisons with state-of-the-art 3D scene generation approaches such as 3DTown. Even though 3DTown’s code is not publicly available, including comparisons with similar open-source or reproduced baselines would make the contribution more convincing.
3. The SDEdit variant section is difficult to follow; the motivation and procedure are not clearly explained. The text should be reorganized and possibly supported with a schematic illustration.
4. The image-conditioned patch generation appears to work primarily with near-top-down viewpoints. For oblique input views (e.g., Figure 7’s first and third examples), significant distortions occur (e.g., warped buildings), which may indicate a limitation of the approach.

**Questions:**

1. Consider expanding the evaluation dataset (e.g., similar to the 100-image benchmark used in 3DTown).
2. Explore or reproduce additional open-source 3D scene generation methods for comparison.
3. Please clarify and reorganize the SDEdit variant explanation, ideally with an accompanying figure to illustrate the workflow.
4. Based on the method description, does Extend3D inherently require near-top-down input images? The distorted outputs for oblique views suggest a potential constraint—can the method handle general perspectives?
5. When patchifying the input image, is super-resolution applied to image patches? If the input image has low resolution, would the resulting 3D scene suffer from loss of geometric or textural detail?

Things to improve the paper that did not impact the score:
- In Figure 1, the notation “a = b = 6” may confuse readers, as its meaning is unclear prior to seeing the figure. Although the abstract mentions these as scaling factors, readers lack intuition about the latent space size—consider removing it.
- In Equation (3), please clarify what $l$ represents; similarly, in Equation (12), explain the operator $\oslash$.
- While Figure 3 serves as a helpful schematic, the equations in Section 4.1 are difficult to follow; consider reorganizing this section for better clarity.
- Line 412: “get” → “got.”
- Please report the computational cost (e.g., inference time or GPU hours) of the proposed method.

---

### Official Review · Reviewer_PsNC · 2025-11-01

**Soundness:** 3
**Presentation:** 1
**Contribution:** 3
**Rating:** 4
**Confidence:** 4

**Summary:**

This paper proposes to leverage Trellis as the 3D generation model to generate  3D town models patch by patch. Given an top-down view image, the proposed method first obtains an initial sparse 3D structure through depth estimation, and then refines the sparse 3D structure with a loss to enforce its consistency with the initialized structure from depth.  Afterwards, it leverages patch-wise rectified flow to generate structured latent for the whole scene.  The designed pipeline is sound and produce impressive experimental results.

**Strengths:**

1. This paper combines SDEdit and Trellis to solve the large scale 3D scene generation problem. While it is limited to top-down view now, it shows the potential of pretrained 3D generation models.
2. This paper did a comprehensive experiments to validate the design choice in the proposed method. It shows how the method works for different types of input images and with the technical components in the ablation studies. The accompanying video demo is impressive.

**Weaknesses:**

1. The symbols used in this paper can be simplified. The are up arrows associated to Z, O, but it seems to be no concrete effect for the equations in this paper.

2. Lack of discussion on the pros and cons of the 3D object generation methods and 3D scene generation methods when applying these two methods to 3D scene generation.  If we train a dedicated 3D scene generation model with estimated depth as an addition condition,  can this model compete with the proposed method?

**Questions:**

regarding the sparse structure initialization part, how is the guidance structure represented?  In the implementation of Trellis, image-conditioned structured sparse structure generation will produce a compressed volumetric latent feature field.  Since the structure initialized from the point cloud is just a occupancy field, we do not know the latent feature. In line 355, Z_o(g) is computed via an VAE encoder? However, since occluded regions are not known at this stage, how do you treat the occluded voxels?  just treat them as non occupied voxels?  I am wondering why not just apply L_SS to guided the rectified flow process and why the guidance structure Z_o(g) is necessary.

---

### Note · Authors · 2025-11-12

**Comment:**

Thank you for your reviews. Based on the comments, we have decided to withdraw our submission. We truly appreciate the constructive suggestions, and we will incorporate them to improve the work for future publication.

**Withdrawal Confirmation:**

I have read and agree with the venue's withdrawal policy on behalf of myself and my co-authors.